# Universal shape and pressure inside bubbles appearing in van der Waals heterostructures

E. Khestanova[1], F. Guinea[1,2], L. Fumagalli[1], A.K. Geim[1] & I.V. Grigorieva[1]

Trapped substances between a two-dimensional (2D) crystal and an atomically flat substrate lead to the formation of bubbles. Their size, shape and internal pressure are determined by the competition between van der Waals attraction of the crystal to the substrate and the elastic energy needed to deform it, allowing to use bubbles to study elastic properties of 2D crystals and conditions of confinement. Using atomic force microscopy, we analysed a variety of bubbles formed by monolayers of graphene, boron nitride and $MoS_2$. Their shapes are found to exhibit universal scaling, in agreement with our analysis based on the theory of elasticity of membranes. We also measured the hydrostatic pressure induced by the confinement, which was found to reach tens of MPa inside submicron bubbles. This agrees with our theory estimates and suggests that for even smaller, sub-10 nm bubbles the pressure can be close to 1 GPa and may modify properties of a trapped material.

[1] School of Physics and Astronomy, University of Manchester, Oxford Road, Manchester M13 9PL, UK. [2] IMDEA Nanociencia, Faraday, 9, Cantoblanco, 28049 Madrid, Spain. Correspondence and requests for materials should be addressed to F.G. (email: francisco.guinea@manchester.ac.uk) or to I.V.G. (email: irina.grigorieva@manchester.ac.uk).

Van der Waals (vdW) heterostructures[1]—stacks of atomically thin layers of different materials assembled layer by layer—are making possible the design of new devices with tailored properties. An essential feature of such heterostructures is atomically clean interfaces that form due to strong adhesion between the constituent layers[2]. Even though the contamination (adsorbed water and hydrocarbons) is inevitably present on individual layers before assembly, the vdW forces that attract adjacent two-dimensional (2D) crystals squeeze out trapped contaminants, usually pushing them into submicron-size 'bubbles' and leaving large interfacial areas atomically sharp and free of contamination[2].

So far such bubbles have been used simply as signatures of good adhesion between constituents of vdW heterostructures and as indicators that the interfacial areas between the bubbles are perfectly clean[3]. Now, we show that the bubbles can be employed as a tool to study the elastic properties of the 2D crystals involved and, also, to evaluate the conditions that nanoscale confinement exerts on the enclosed material (for example, hydrostatic pressure). This information is important in many situations, where confinement can modify materials properties, with water inside graphene nanocapillaries[4–6], nanocrystals or biological molecules confined in graphene liquid cells[7–10], room-temperature ice in a 2D nanochannel[11,12] and a hydrothermal anvil made of graphene on diamond[13] being a few examples. Furthermore, highly strained graphene nano-bubbles have been shown to possess enormous pseudo-magnetic fields[14], $>300$ T. The detailed knowledge of strain for commonly occurring bubbles should facilitate studies of the electronic properties of graphene under conditions inaccessible in high-field magnet laboratories[15].

Here we study bubbles formed between a 2D crystal (monolayer graphene, monolayer hexagonal boron nitride (hBN) or monolayer $MoS_2$) and an atomically smooth flat substrate (hBN, graphite and $MoS_2$). By analysing shapes and dimensions of the bubbles, and comparing them with the corresponding predictions of the elasticity theory, we find that the bubbles for all three materials are fully described by the combination of a 2D crystal's elastic properties and its vdW attraction to a substrate. We find excellent agreement between the experiment and theory, both for smoothly deformed bubbles, and for bubbles with shape and dimensions modified by a residual strain. Furthermore, using indentation of bubbles with an atomic force microscope (AFM) tip, we extracted the hydrostatic (vdW) pressure inside them, and Young's moduli for graphene and $MoS_2$ membranes. Through the experiments and analysis below, we found that in-plane stiffness of 2D crystals plays a major role in determining characteristic shapes and density of the bubbles one can expect to find when such a crystal is part of a vdW heterostructure. Stiffer 2D crystals, such as graphene or monolayer hBN on an hBN substrate, form smaller, more sparsely distributed bubbles, so that large (up to $100\,\mu m^2$) areas of the structure present a perfect vdW interface. This has been exploited in fabrication of high-quality electronic devices. On the other hand, stronger adhesion between a 2D crystal and the substrate (monolayer $MoS_2$ on an $MoS_2$ substrate being an example) can be exploited to achieve a higher vdW pressure inside the bubbles, which is desirable if one wants to modify the properties of a material through nanoscale confinement.

## Results

**Experiment**. Samples for this study were made by mechanical exfoliation of graphene, hBN and $MoS_2$ monolayers onto hBN, graphite and $MoS_2$ substrates using the now standard dry-peel technique[3,16]. To this end graphene/monolayer hBN/monolayer

$MoS_2$ were first mechanically exfoliated onto a poly(methyl methacrylate) membrane. The latter was then loaded into a micromanipulator, where it was placed face-down onto a substrate (a $\sim100$ nm thick crystal of graphite, hBN or $MoS_2$ on a $Si/SiO_x$ wafer), after which the supporting polymer membrane was mechanically peeled off, ensuring residue-free surface of a 2D crystal. The resulting heterostructures were then heated (annealed[3,16]) at 150 °C for 20–30 min, which resulted in spontaneous formation of a large number of bubbles filled with hydrocarbons[2], with typical separations from $\sim0.5$ to tens of microns. The annealing time and temperature were optimized to ensure that the bubbles reached equilibrium conditions, that is, no further changes in their shape, size or position could be detected with further annealing. After that the dimensions and topography of many bubbles (up to 100 for each heterostructure) were analysed using AFM.

Figure 1 shows typical examples of bubbles formed by monolayer graphene on bulk hBN. The majority of the bubbles were $<500$ nm in radius, $R$, and had a round or nearly round base (Fig. 1a). Larger bubbles typically exhibited pyramidal shapes, with either triangular (Fig. 1b) or trapezoidal (Fig. 1c) bases. Bubbles formed by monolayer hBN on bulk hBN were also either round or approximately triangular in shape, but smaller in size compared with graphene ($<100$ nm for round and $<500$ nm for triangular bases). Bubbles formed by $MoS_2$ monolayers were mostly round, similar to those shown in Fig. 1a for graphene, but exhibited a broader size distribution, with $30<R<1,000$ nm. We measured the cross-sectional profiles of the observed bubbles and analysed their maximum height, $h_{max}$, and the aspect ratio of $h_{max}$ to the radius, $R$, or to the length of the side, $L$, as appropriate.

The results for round-type graphene bubbles are shown in Fig. 2a. The aspect ratio, $h_{max}/R$, is remarkably universal, that is, independent of the bubbles' radius, $R$, or volume, $V$: $h_{max}/R \approx 0.11$, within $\sim10\%$. Moreover, if we discount the smallest bubbles with $R<50$ nm, the accuracy reaches 4% for sizes varying by an order of magnitude. Very similar behaviour was found for monolayer hBN, with $h_{max}/R \approx 0.11$ for bubbles $>50$ nm and a somewhat increasing $h_{max}/R$ for $R<50$ nm—see Fig. 2a. Only a few sufficiently large bubbles were found in this case, limiting our analysis.

Aspect ratios for round bubbles formed by $MoS_2$ monolayers are shown in Fig. 2b. For comparison, we analysed the $MoS_2$ bubbles formed on two different substrates, $MoS_2$ and hBN. Again, for the same 2D crystal–substrate combination we find a constant $h_{max}/R$, but its value depends on the substrate and is notably larger compared with graphene and hBN monolayers. This can be attributed to different elastic properties of monolayer

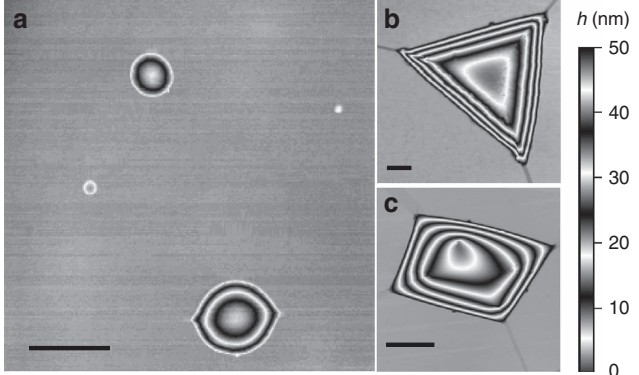

**Figure 1 | Graphene bubbles.** (**a**–**c**) AFM images of graphene bubbles of different shapes. Scale bars, 500 nm (**a**); 100 nm (**b**); 500 nm (**c**). The vertical scale on the right indicates the height of the bubbles.

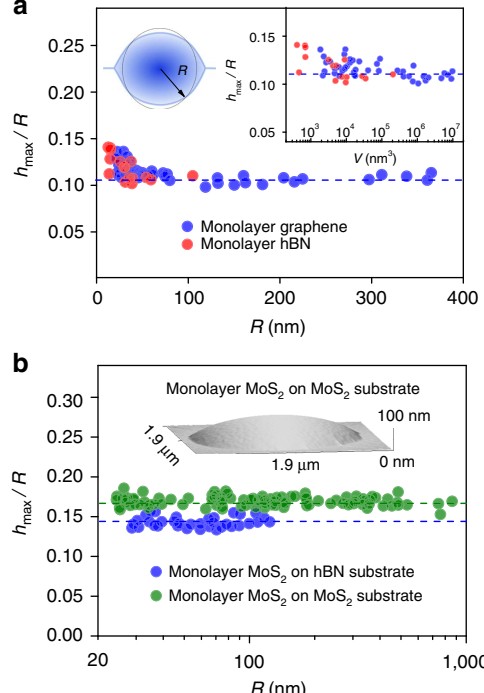

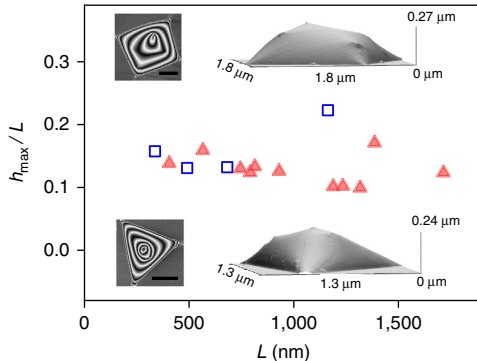

**Figure 4 | Aspect ratio of pyramidal graphene bubbles with sharp features.** Symbols show the measured aspect ratios of triangular (red-closed symbols) and trapezoidal (blue-open) graphene bubbles on hBN substrates, as a function of their side length, $L$, determined as $L=\sqrt{4A/\sqrt{3}}$ for triangular bubbles and as $L=\sqrt{A}$ for trapezoidal ones. Here, $A$ is the measured area of the base of a bubble. Insets show typical AFM images of such bubbles: left and right are top and 3D views, respectively. Scale bars, 500 nm.

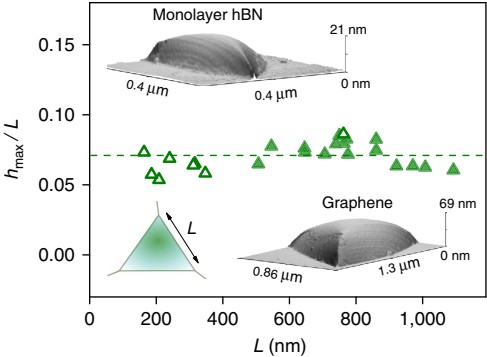

**Figure 2 | Universal shape of round-type bubbles.** (**a**) Measured aspect ratios as a function of the base radius for graphene (blue symbols) and monolayer hBN (red symbols). Dashed line shows the mean value. Top left inset: sketch of a nearly round bubble and its effective radius $R$ determined as $R=\sqrt{A/\pi}$, where $A$ is the measured area of the base of the bubble. Right inset: aspect ratio of the bubbles as a function of their volume. (**b**) Aspect ratio of $MoS_2$ bubbles on hBN and $MoS_2$ substrates. Dashed lines show the mean values of $h_{max}/R = 0.14$ and 0.17, respectively. The logarithmic scale is used to accommodate the large range of $R$. Inset: AFM image of a typical $MoS_2$ bubble.

**Figure 3 | Aspect ratio of smooth triangular bubbles.** Symbols show the measured aspect ratios of graphene and hBN bubbles (closed and open symbols, respectively), both on hBN substrates, as a function of $L$. The dashed line shows the mean aspect ratio, $h_{max}/L = 0.07$. Bottom left inset: sketch of a triangular bubble. Its side length $L$ was experimentally determined as $L=\sqrt{4A/\sqrt{3}}$, where $A$ is the measured area of the base of a bubble. The other two insets show typical AFM images of smoothly deformed triangular bubbles.

$MoS_2$ compared with one-atom-thick crystals (graphene and monolayer hBN). Furthermore, the different $h_{max}/R$ found for different substrates point at the importance of vdW adhesion, as discussed below.

A constant aspect ratio was also found for graphene bubbles with triangular bases, such as those shown in Figs 1b and 3. In

this case, it is intuitive to use the length of the side, $L$, to characterize their sizes. Similar to the round bubbles in Fig. 2a, these bubbles usually had smooth round tops, but were larger in size (typical $L$ between 500 and 1,000 nm) and exhibited the aspect ratio $h_{max}/L = 0.07 \pm 0.01$—see Fig. 3. We note that, although this value appears to be lower than that for the round bubbles, as if the triangular bubbles were somewhat thinner, this is simply the effect of using a different measure to characterize the lateral size ($L$ versus $R$). Indeed, redefining the lateral size of triangular-type bubbles as a distance $L^{\star}$ from their centres to corners, we find the same ratio $h_{max}/L^{\star}$ as for round bubbles, within our experimental accuracy. As discussed below, the shapes and dimensions of all smoothly deformed bubbles (round or triangular) are expected to follow the same scaling.

The only class of bubbles that showed strong deviations from the universal scaling behaviour were pyramidal-type bubbles with sharp features. They exhibited sharp ridges that often extended nearly to the full height of the bubbles. Two examples are shown as insets in Fig. 4. The aspect ratio, $h_{max}/L$, for such bubbles showed relatively large variations (by a factor of 2), with most values being higher than those for smoothly deformed bubbles—c.f. Figs 3 and 4.

To summarize, all bubbles—formed by graphene, hBN and $MoS_2$ monolayers—exhibited a small set of shapes (mostly, round and triangular) with a universal aspect ratio. Different shapes were found on each sample, but the frequency of occurrence was different for different shapes, and there was a correlation between the bubbles' shapes and their sizes. For example, all possible bubble shapes (round, triangular and pyramidal) were found on the same sample of graphene on an hBN substrate. Of these, bubbles with $R < 400$ nm were round or nearly round and most of them were $< 200$ nm; bubbles with $500 < R < 1,000$ nm were triangular with smooth tops, and triangular and pyramidal bubbles with sharp features were very few, with a broad distribution of sizes, from 400 to 1,400 nm. For monolayer hBN, bubbles of all shapes tended to be smaller; accordingly, the size ranges were different (20–100 nm for round bubbles; 150–350 nm for triangular with smooth tops), but a correlation between shape and size was found as well. These statistics are summarized in Supplementary Fig. 1.

In terms of the aspect ratio, monolayers of graphene and hBN, which have similar elastic properties, showed the same aspect ratio. The aspect ratio for $MoS_2$, which has a lower elastic

**Figure 5 | Sketch of the bubble considered in our theoretical analysis.** The bubble is formed by material trapped between a substrate and a 2D layer (graphene).

stiffness[17,18], was also constant, but its value was up to 50% higher than for graphene and hBN monolayers. The universal behaviour for different 2D crystals points to the definitive role played by their elastic properties, as analysed in the following sections.

**Scaling analysis.** To model the observed bubbles, we consider a material trapped between a flat substrate and a 2D crystal attracted to the substrate by vdW forces (Fig. 5). We note that a related situation—circular gas-filled graphene bubbles on a substrate—was analysed recently using nonlinear elastic plate and membrane theory[19], and numerical simulations[20]. However, the results of refs 19,20 are not applicable to our experiments because they considered bubbles under constant pressure with clamped edges. In contrast, our theory corresponds to the problem studied experimentally, that is, bubbles of a constant volume, where the edges adapt to the competition between the vdW attraction and the internal pressure, while the pressure itself is determined by the adhesion between the 2D crystal and the substrate. Furthermore, the 2D crystal is free to adapt to the substrate and the bubble profiles are not assumed (as in ref. 19), but found self-consistently.

For simplicity, below we refer to graphene only. Its rigidity is determined by a combination of the in-plane stiffness, and the energy associated with out-of-plane bending. The in-plane stiffness is described by the theory of elasticity[21], which requires the specification of two parameters, Young's modulus, $Y$, and Poisson's ratio, $v$, or, alternatively, Lamé coefficients, $\lambda$ and $\mu$. As graphene is an ultimately thin 2D membrane, out-of-plane deformations lead to in-plane stresses, making the system highly anharmonic[22]. The out-of-plane bending is described by the bending rigidity, $\kappa$. Relative contributions of the in-plane stiffness and the bending rigidity to the elastic energy of a 2D membrane are determined by the scale of deformations: beyond a length scale $\ell_{anh} \sim \sqrt{Y/\kappa}$ the stiffness is dominated by in-plane stresses. For graphene, this scale is $\ell_{anh} \approx 4\,\text{Å}$, so that in most situations the bending rigidity can be neglected (however, see further). The equivalent length for $MoS_2$ is somewhat larger, but still $<1\,\text{nm}$.

The vdW energy associated with separating of a graphene layer from the substrate is given by

$$E_{vdW} = \pi\gamma R^2$$
$$\gamma = \gamma_{GS} - \gamma_{Gb} - \gamma_{Sb} \tag{1}$$

where $\gamma_{GS}$, $\gamma_{Gb}$ and $\gamma_{Sb}$ are the adhesion energies between graphene and the substrate, graphene and the substance inside the bubble, and the substrate and the substance, respectively.

If the bubble is filled with a substance having a finite compressibility, $\beta$, it can be written as

$$\beta^{-1} = V\frac{\partial^2 E_b(V)}{\partial V^2} = -V\frac{\partial P}{\partial V} \tag{2}$$

where $E_b(V)$ is the free energy of the substance inside the bubble of volume $V$ and $P$ is the pressure.

The bubble's height profile is described by

$$h(r) = h_{max}\tilde{h}\left(\frac{r}{R}\right) \tag{3}$$

where $h_{max}$ is the maximum height of the bubble, so that $\tilde{h}(0)=1, \tilde{h}(1)=0$. The in-plane displacements are defined by the function $u_r(r)=(h_{max}^2/R)\times\tilde{u}_r(R)$. We assume radial symmetry, so that the azimuthal displacements vanish, that is, $u_\theta = 0$. Details of calculating the in-plane displacements and the total energy as a function of $h(r)$ are given in Supplementary Note 1.

Neglecting the bending rigidity, the total energy can be written as

$$E_{tot} = E_{el} + E_{vdW} + E_b(V) =$$
$$= c_1\left[\tilde{h}\right]Y\frac{h_{max}^4}{R^2} + c_2\left[\tilde{h}\right]Y\epsilon h_{max}^2 + \pi\gamma R^2 + E_b(V) \tag{4}$$

where dimensionless coefficients $c_1$ and $c_2$ depend only on the function $\tilde{h}$, describing the height profile, and the volume $V$ is

$$V = c_V\left[\tilde{h}\right]h_{max}\times R^2. \tag{5}$$

Below, we show that the function $\tilde{h}(x)$ is generic, that is, independent of the material parameters $Y$, $\gamma$ and $E_b(V)$.

By minimizing Equation (4) with respect to $h_{max}$ and $R$, we obtain

$$c_1Y\frac{4h_{max}^3}{R^2} + 2c_2Y\epsilon h_{max} - c_VR^2P = 0$$
$$-c_1Y\frac{2h_{max}^4}{R^3} + 2\pi\gamma R - 2c_Vh_{max}RP = 0, \tag{6}$$

where we have used $P = -\partial E_b/\partial V$. By eliminating $P$ in Equation (6), we obtain

$$5c_1Y\left(\frac{h_{max}}{R}\right)^4 + 2c_2Y\epsilon\left(\frac{h_{max}}{R}\right)^2 - \pi\gamma = 0 \tag{7}$$

This equation defines the aspect ratio of the bubble, $h_{max}/R$, in terms of the coefficients $c_1$ and $c_2$, parameters $Y$ and $\gamma$, and an external strain, $\epsilon$:

$$\left(\frac{h_{max}}{R}\right)^2 = -\frac{c_2\epsilon}{5c_1} + \sqrt{\left(\frac{c_2\epsilon}{5c_1}\right)^2 + \frac{\pi\gamma}{5c_1Y}} \tag{8}$$

In the absence of external strain, $\epsilon = 0$, this expression reduces to

$$\frac{h_{max}}{R} = \left(\frac{\pi\gamma}{5c_1Y}\right)^{1/4} \tag{9}$$

that is, the value of $h_{max}/R$ is determined solely by the balance between vdW and elastic energies of a 2D crystal, independent of the properties of the substance captured within the bubble. This result is in excellent agreement with the constant aspect ratios observed experimentally—see Figs 2 and 3.

The presence of finite $\epsilon$ (induced, for example, during fabrication) should modify the bubbles' shape, reducing the aspect ratio $h_{max}/R$ for tensile strains and increasing it for compressive strains—see Supplementary Notes 2 and 3.

The above analysis also shows that the fluid material inside the bubble is under a constant hydrostatic pressure $P$, which is described by Equation (6) and, following ref. 12, is referred to below as vdW pressure. Accordingly, our case of bubbles formed by the competition of vdW and elastic forces can be considered as a particular case of the membrane deformed by applying a constant external pressure.

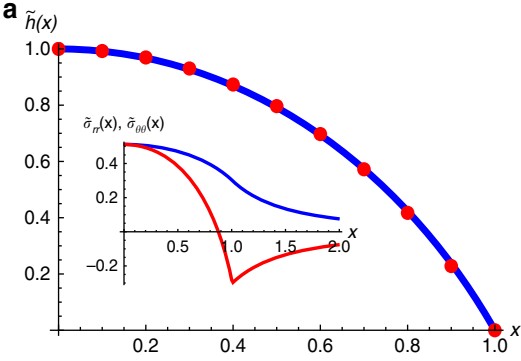

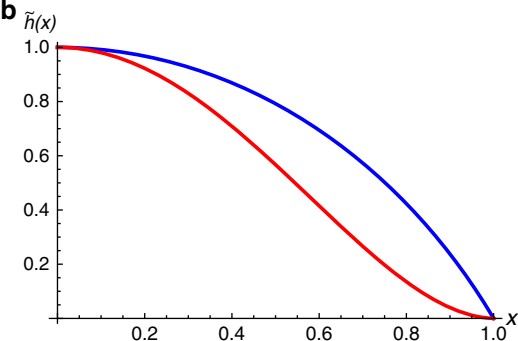

**Figure 6 | Bubble profiles.** (**a**) Scaling function, $\tilde{h}(x)$, obtained by minimizing numerically the elastic energy. It is well approximated by a quartic function, $\tilde{h}(x) = 1 - x^2 + c(x^2 - x^4)$ (red dots), with $c = 0.25$. The inset shows the scaled stresses, $\tilde{\sigma}_{rr}(x)$ and $\tilde{\sigma}_{\theta\theta}(x)$ (blue and red curves, respectively). (**b**) Comparison between the bubble profiles under a hydrostatic pressure for the cases dominated by in-plane strains (blue) and bending (red).

Using the change of variables, $x = r/R$, we write the total energy as

$$
\begin{aligned}
E_{tot} &= E_{el} + E_{bend} + P \times V \\
E_{el} &= c_1\left[\tilde{h}(x)\right] Y \frac{h_{max}^4}{R^2} + c_2\left[\tilde{h}(x)\right] Y \epsilon h_{max}^2 \\
E_{bend} &= c_3\left[\tilde{h}(x)\right] \kappa \frac{h_{max}^2}{R^2} \\
E_P &= c_V\left[\tilde{h}(x)\right] P h_{max} R^2
\end{aligned}
\tag{10}
$$

We consider first the bubble's profile, $\tilde{h}(x)$, determined solely by the competition between the pressure and the in-plane stresses, $E_{el}$ and $E_P$, and we set $\epsilon = 0$. Minimization of $E_{tot}$ with respect to $h_{max}$ gives

$$
\begin{aligned}
h_{max} &= \left[\frac{c_V(\tilde{h})}{c_1(\tilde{h})}\right]^{1/3} \left(\frac{PR^4}{4Y}\right)^{1/3} \\
E_{tot}\left[\tilde{h}\right] &= -\frac{3}{4}\left[\frac{c_V^4(\tilde{h})}{c_1(\tilde{h})}\right]^{1/3} \left(\frac{P^4 R^{10}}{4Y}\right)^{1/3}
\end{aligned}
\tag{11}
$$

We can now calculate $\tilde{h}$ by minimizing $E_{tot}$. This yields that $\tilde{h}(x)$ is universal, that is, independent of $Y$, $P$ and $R$. The function $\tilde{h}(x)$ is shown in Fig. 6. The in-plane stresses associated with the bubble formation can also be expressed in a scaled form, $\tilde{\sigma}_{rr}(x) = \left[R^4/\left(h_{max}^4 Y\right)\right] \times \sigma_{rr}(r/R)$ and $\tilde{\sigma}_{\theta\theta}(x) = \left[R^4/\left(h_{max}^4 Y\right)\right] \times \sigma_\theta(r/R)$. These functions are plotted in the inset of Fig. 6. It is interesting to note that the hoop stress, $\sigma_{\theta\theta}$, becomes negative (compressive) near the base of the bubble. In the absence of vdW pressure, a compressive stress can lead to an instability with respect to the formation of wrinkles[23]. The existence of in-plane stresses outside the bubble (see inset in Fig. 6) implies that the

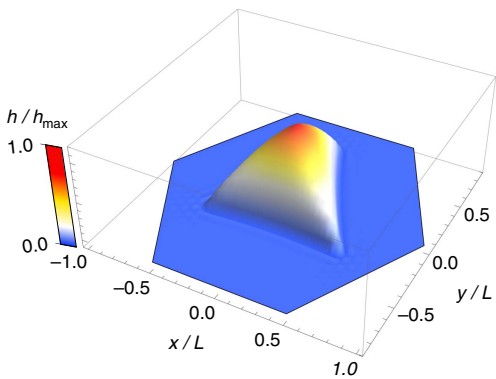

**Figure 7 | Universal shape of smooth triangular bubbles (theory).** Shown is the result of numerical calculations of the 3D shape of a triangular bubble with an equilateral triangle as its base. The bubble dimensions are scaled to its maximum height, $h_{max}$, and the side length, $L$.

bubbles interact with each other—see Supplementary Note 4, where this interaction is analysed. It is attractive and decays as $Y/d^2$, where $d$ is the distance between bubbles.

A similar analysis can be carried out when the shape of the bubble is determined by the bending rigidity, and $E_{tot} = E_{bend} + E_P$. In this case, we find

$$
\begin{aligned}
h_{max} &= \frac{c_V(\tilde{h})}{2c_3(\tilde{h})} \frac{PR^4}{\kappa} \\
E_{tot}\left[\tilde{h}\right] &= -\frac{c_V(\tilde{h})}{4c_3(\tilde{h})} \frac{P^2 R^6}{\kappa}
\end{aligned}
\tag{12}
$$

The generic profiles in the two cases (elastic energy is dominated by either in-plane stresses or bending) are given in Fig. 6.

In the following, we neglect the bending rigidity term, $E_{bend}$, as appropriate for 2D membranes with $\sqrt{\kappa/Y} \ll h_{max}, R$, and corresponds to the case studied in our experiments. In principle, the coefficients $c_1$ and $c_2$, and the function $\tilde{h}$ can depend on strain $\epsilon$. However, we have found numerically that this dependence is negligible for $\epsilon \lesssim 0.1$, that is, can be neglected in realistic situations because even smaller strains (a few %) are likely to cause slippage along the substrate due to limited adhesion. The numerical parameters that relate $h_{max}$, $L$, $Y$ and $P$ are found as

$$
\begin{aligned}
c_1 &\approx 0.7 \\
c_2 &\approx 0.6 \\
c_V &\approx 1.7.
\end{aligned}
\tag{13}
$$

The above scaling analysis can also be applied to bubbles of other shapes, such as the pyramidal bubbles found experimentally (Fig. 3). For simplicity, we model smooth triangular bubbles as having an equilateral triangle as their base. The bubbles are then characterized by two length scales: height, $h_{max}$, and the side length, $L$. The scaled universal profile for a triangular bubble is shown in Fig. 7. The numerical parameters in this case are

$$
\begin{aligned}
c_1 &\approx 0.6 \\
c_2 &\approx 0.3 \\
c_V &\approx 0.2.
\end{aligned}
\tag{14}
$$

The corresponding average strain, $\bar{u}_{rr}$, for graphene/hBN/MoS$_2$ monolayers enclosing a bubble is of order

$$
\bar{u}_{rr} \approx \left(\frac{h_{max}}{R}\right)^2 \approx 1-2\ \%
\tag{15}
$$

To gain further insight, we estimate the parameters in Equation (9), corresponding to the experimentally observed

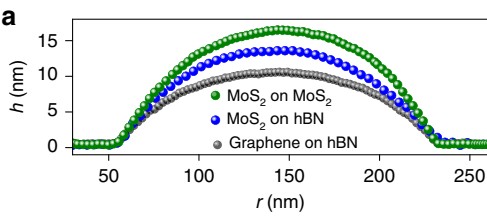

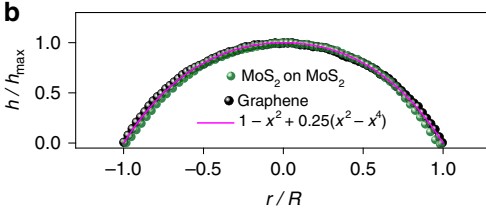

**Figure 8 | Universal profile of round bubbles.** Scaled and normalized cross-sectional profiles measured experimentally for typical round bubbles formed by graphene and $MoS_2$ monolayers. (**a**) Comparison of the profiles of graphene and $MoS_2$ bubbles scaled to the same radius, $R$, to emphasize their different heights. The actual sizes are $R = 168$ nm for graphene, and $R = 97$ and 816 nm for $MoS_2$ monolayers on hBN and $MoS_2$, respectively. (**b**) Same data but scaled in both dimensions. The symbols show the experimental profiles; the solid curve is the theory profile as in Fig. 6a, with no fitting parameters.

aspect ratio $h_{max}/R \approx 0.11$ for circular and $h_{max}/L \approx 0.07$ for triangular graphene bubbles. Using the known stiffness of graphene, $Y_G \approx 22$ eV $Å^{-2}$, this yields an effective adhesion energy $\gamma \sim 0.005$ eV $Å^{-2}$, significantly lower than the measured value for adhesion between graphene on $SiO_x$, $\sim 0.03$ eV $Å^{-2}$ (ref. 24) and, also, lower than the vdW adhesion found theoretically[25,26], $\sim 0.01–0.02$ eV $Å^{-2}$. This indicates that the adhesion between graphene (or hBN and $MoS_2$) and the trapped hydrocarbon contamination, $\gamma_{Gb}$, is comparable to that between graphene and the substrate, $\gamma_{GS}$, as expected for these lipophilic 2D crystals. Note that $\gamma_{Gb}$ should be smaller than $\gamma_{GS}$. Otherwise, no bubbles would be formed as the contaminating materials would tend to spread along the substrate. As we show below, a similarly low $\gamma$ follows from our AFM measurements of vdW pressure inside bubbles (see the 'Pressure inside the bubbles' section).

hBN has approximately the same stiffness as graphene[27] that results in similar aspect ratios because they depend only weakly of $Y$ (as $Y^{1/4}$). On the other hand, $MoS_2$ is significantly less stiff, with twice lower Young's modulus $Y_{MoS_2} \approx 11.2$ eV $Å^{-2}$ (refs 17,18,28). This translates into a larger aspect ratio compared with graphene and hBN, in agreement with the experiment. Furthermore, notably different aspect ratios for $MoS_2$ bubbles on hBN and $MoS_2$ substrates ($\approx 0.14$ versus $\approx 0.17$; Fig. 2b) can be attributed to different $\gamma$ for the two substrates; see Equation (9).

Figure 8 compares the calculated universal profile, $\tilde{h}(x)$, with those observed experimentally for the round bubbles formed by graphene and $MoS_2$ monolayers. In both cases, the profiles are remarkably well described by the quartic function shown in Fig. 6a. This proves that not only the aspect ratio, $h_{max}/R$, but also the shape of the bubbles is universal and determined solely by the elastic properties of 2D crystals and their adhesion, independent of the properties of the trapped material.

**Deviations from scaling.** In the experiments, the predicted scaling behaviour breaks down for large pyramidal bubbles with sharp ridges, pointed summits and relatively flat facets, such as those shown in Fig. 4. They exhibit a significant spread of $h_{max}/L$

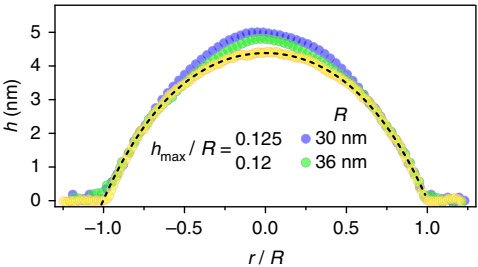

**Figure 9 | Deviations from the universal profile in small bubbles.** Scaled experimental profiles of small graphene bubbles showing deviations (blue and green symbols). For comparison, the universal profile of a larger graphene bubble is shown by yellow symbols, together with the fit to the theoretical scaling function (dashed curve).

values, from $\sim 0.09$ to 0.2, which are also larger than the values found for smooth bubbles (Fig. 3).

Sharp ridges between the flat facets minimize the in-plane elastic energy at the cost of bending along the length of the ridge[29]. Therefore, we assume that most of the elastic energy of such bubbles resides in the ridges. Following the analysis in ref. 29, we consider a ridge of length $L$ separating two flat facets that make an angle $\theta$. For a bubble of height $h_{max}$ with a pyramidal shape, where the base is a polygon with the side length $L$, we have $\theta \approx h_{max}/L$. At the centre of the ridge, its curvature can be described by the radius $R_{ridge}$ and, to allow for that curvature to exist, the ridge has to sag by an amount $\xi \sim R_{ridge}\theta^2$. The strained area around the ridge has a width $w \sim R_{ridge}\theta$ and a length $L$. Then, the resulting in-plane elastic energy is of the order of

$$E_{str} \approx YwL\left(\frac{\xi}{L}\right)^2 \approx Y\frac{R_{ridge}^5}{L^3}\theta^9 \qquad (16)$$

The associated bending energy scales as

$$E_{bending} \approx \kappa\frac{wL}{R_{ridge}^4} \approx \kappa\frac{L}{R_{ridge}}\theta \qquad (17)$$

The optimal value of $R_{ridge}$ makes these two energies comparable, so that

$$R_{ridge} \approx \left(\frac{L}{\theta^2}\right)^{2/3}\left(\frac{\kappa}{Y}\right)^{1/6} \qquad (18)$$

and the total elastic energy is of the order of

$$E_{str} + E_{bending} \approx \kappa L^{1/3}\theta^{5/3} \approx \kappa\left(\frac{Y}{\kappa}\right)^{1/6}\frac{h_{max}^{5/3}}{L^{4/3}}\left(\frac{Y}{\kappa}\right)^{1/6}. \qquad (19)$$

The relation between $h_{max}$ and $L$ is given by the minimization of the elastic and vdW energies, where, as in the previous section, $E_{vdW} \propto \gamma L^2$. We finally find

$$\frac{h_{max}^{5/3}}{L^{10/3}} \propto \frac{\gamma}{\kappa^{5/6}Y^{1/6}}$$
$$\frac{h_{max}}{L} \propto L\frac{\gamma^{3/5}}{\kappa^{1/2}Y^{1/10}} = \frac{L}{L_0} \qquad (20)$$

with

$$L_0 = \frac{\kappa^{1/2}Y^{1/10}}{\gamma^{3/5}}, \qquad (21)$$

that is, the aspect ratio of bubbles with sharp ridges is not constant, but depends on the size and geometry of the bubbles, which explains the absence of a universal scaling in this case as observed experimentally.

Deviations from the universal profile were also found for very small graphene and hBN bubbles, $R \lesssim 50$ nm, despite the fact that

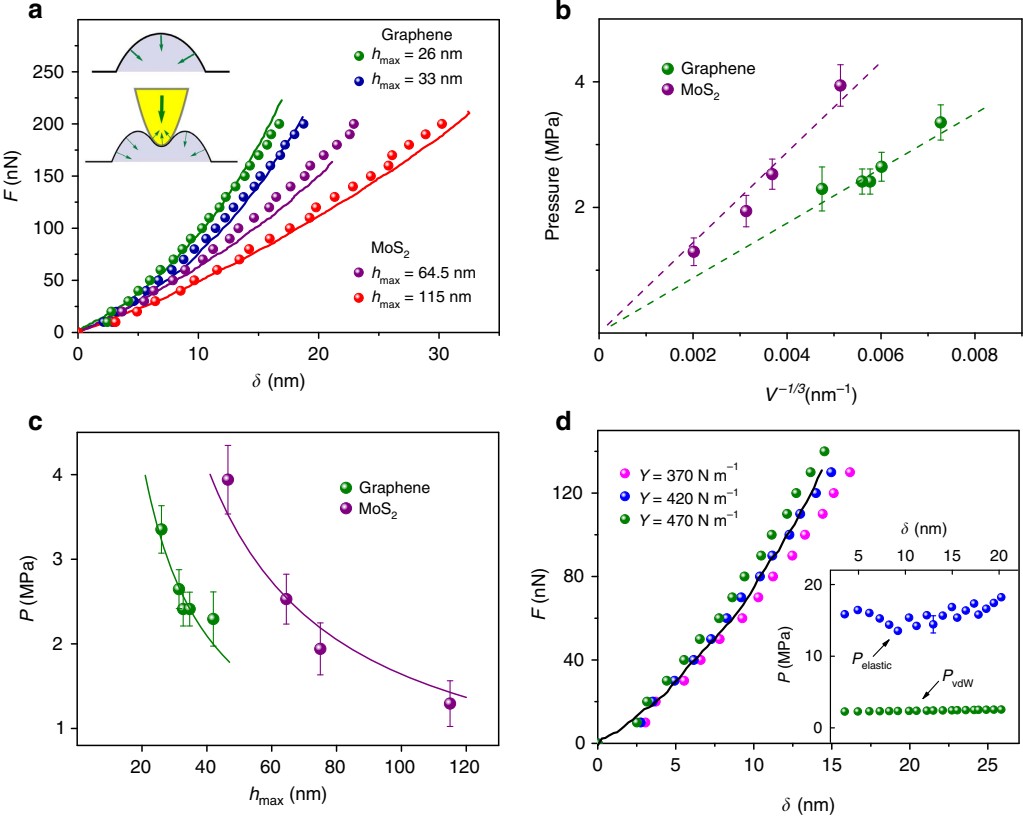

**Figure 10 | Measuring vdW pressure.** (**a**) Experimental force-indentation curves (solid lines) and numerical fits (symbols) for two graphene bubbles and two monolayer $MoS_2$ bubbles of different sizes (for graphene bubbles $h_{max} = 26$ and 33 nm, $R = 240$ and 320 nm; and for $MoS_2$ bubbles $h_{max} = 64.5$ and 115 nm, $R = 437$ and 800 nm, respectively). Inset: sketch of the forces exerted on an AFM tip and the material inside the bubble before (top) and during (bottom) indentation. (**b**) vdW pressure in graphene and $MoS_2$ bubbles extracted from theory fits shown in **a** as a function of the bubbles' volume. Dashed lines are linear fits to the data. (**c**) Same pressure data but as a function of the bubbles' maximum height. Solid lines are fits to $\propto 1/h_{max}$. (**d**) An experimental force-indentation curve (black curve) and corresponding numerical fits for three different values of Young's modulus. Inset: comparison of the typical values of pressure exerted on the AFM tip due to the elastic deformation of graphene, $P_{elastic}$, and of the vdW pressure inside the bubble, $P_{vdW}$. Shown are data for one of the graphene bubbles ($h_{max} = 33$ nm and $R = 320$ nm).

they seem to be smooth and almost perfectly round (Fig. 2a). We attribute the breakdown of scaling in this case to residual strain in the 2D layer, that is, unlike large round bubbles, small ones were not fully relaxed during their annealing. This is consistent with the bubbles' profiles observed in the two cases—see Fig. 9. While all larger bubbles exhibited the universal profile described by $\tilde{h}(x) = 1 - x^2 + c(x^2 - x^4)$ (yellow dots in Fig. 9), the small ones showed notable deviations near the top (blue and green dots in Fig. 9), indicating some residual compressive strain. The latter favours higher values of $h_{max}/R$; see Supplementary Note 3. Using this Supplementary Note's equations, we estimate that the observed deviations in small bubbles would require a compressive strain $|\epsilon|$ of the order of $\lesssim 10^{-3}$, in good agreement with remnant strains usually observed by Raman spectroscopy[30,31].

The above analysis allows us to draw important conclusions about the shapes and sizes of the bubbles versus the rigidity of 2D membranes and their adhesion to the substrates. The highest order term in the bubble's energy, $\propto h_{max}^4/R^2$ (Equation (4)) is the elastic energy due to the deformation of the 2D crystal, proportional to the in-plane stiffness, $Y$. Therefore, on average, softer membranes, such as monolayer $MoS_2$, form more bubbles compared with, for example, graphene (cf. Supplementary Fig. 1a,c) and most of them have round or nearly round bases. Large bubbles made by stiffer graphene tend to be either triangular or pyramidal because formation of ridges associated

with such shapes allows some relaxation of strain and therefore reduction in elastic energy. Accordingly, all graphene bubbles with $R > 350$ nm are either triangular or pyramidal, in contrast to $MoS_2$ bubbles of similar size that are (nearly) round.

Adhesion energy appears to play a smaller role in bubble shapes. Nevertheless, bubbles formed by a relatively soft membrane, such as $MoS_2$, are notably higher on a substrate to which they have greater adhesion ($MoS_2$ on $MoS_2$) due to the associated higher pressure inside the bubbles (Equation (11)).

Finally, there is a correlation between the shape and size of the bubbles, and the presence of any other stresses in the 2D membranes, for example, in the vicinity of folds or atomic-scale steps on the substrate. As stresses associated with each bubble are not limited to its visible raised part, but extend to distances $> 2R$ from the bubble centre (Fig. 6a), larger bubbles are attracted to areas of stress concentration and therefore found more often near folds, wrinkles or steps.

**Pressure inside the bubbles.** Equation (6) allows us to calculate the pressure inside a bubble:

$$P = \frac{Y}{c_V h_{max}} \left[ 4c_1 \left( \frac{h_{max}}{R} \right)^4 + 2c_2 \epsilon \left( \frac{h_{max}}{R} \right)^2 \right] \qquad (22)$$

Using Equations (8) and (9), we find, for $\epsilon = 0$,

$$P = \frac{4\pi\gamma}{5c_V h_{\max}}, \tag{23}$$

that is, vdW pressure is determined by the adhesion between a 2D crystal and the substrate and their separation. This expression is similar to the estimate given in ref. 12.

To find the dependence of $P$ on the bubble volume, $V$, we write $h_{\max}$ as a function of $V$ and obtain

$$P = \frac{4\pi\gamma}{5Cv}\left(\frac{5c_1 Y}{\pi\gamma}\right)^{1/6}\left(\frac{c_V}{V}\right)^{1/3} \tag{24}$$

The pressure is independent of the compressibility of the material within the bubble, that is, $V$ adjusts itself in such a way that the pressure exerted on the material inside it (or, vice versa, acting on a 2D membrane) has the value required by the equilibrium between vdW and elastic forces.

If a gas is trapped inside, its compressibility depends on temperature, and, for a monoatomic gas, $P = Nk_B T/V$. This relation, together with Equation (9), implies

$$h_{\max} = \left(\frac{Nk_B T}{c_V P}\right)^{1/3}\left(\frac{\pi\gamma}{5c_1 Y}\right)^{1/6} \tag{25}$$

and, combined with Equation (23), gives

$$P = \left(\frac{4\pi\gamma}{5Cv}\right)^{3/2}\left(\frac{c_V}{Nk_B T}\right)^{1/2}\left(\frac{5c_1 Y}{\pi\gamma}\right)^{1/4} \tag{26}$$

For a $1\,\mu m^3$ volume of a gas captured under ambient conditions (1 atm at room $T$) between a substrate and a 2D membrane, the gas would be compressed to $\sim 1\%$ of its initial volume and experience $P$ of $\approx 4\,MPa$. This implies a density of $2 \times 10^{20}\,cm^{-3}$, which is likely to turn many gases (including water vapour) into liquids, and the preceding analysis (Equations (22–24)) then becomes more appropriate.

To measure the pressure inside the observed bubbles, we used nanoindentation with an AFM tip, an approach similar to that used, for example, in ref. 32 to measure the osmotic pressure inside viral particles, or suggested in ref. 33 for indentation of pressurized elastic shells of finite thickness. We indented bubbles of different sizes with an AFM tip and recorded their force-displacement curves (FDCs; see Methods for details). To ensure a smooth spherical shape of the used AFM tips, they were annealed at a high temperature (Supplementary Fig. 2). Typical FDC's for several graphene and $MoS_2$ bubbles are shown in Fig. 10a. One can see that, as the bubble size decreases, the force, $F$, required to achieve a certain indentation depth, $\delta$, increases. This is qualitatively consistent with the expectation that the vdW pressure should increase as $1/h_{\max}$ (Equations (23) and (24)). However, the force measured by a nanoindentation probe results not only from the resistance due to a finite pressure inside a bubble, but also from the accompanying elastic deformation of the 2D membrane, as we show next.

To separate these two contributions, we have analysed the total energy of a pressurized bubble subject to indentation from an approximately spherical AFM tip. The forces acting on the enclosed material and on the AFM tip are sketched in the inset of Fig. 10a. In the absence of indentation, graphene induces a downward pressure described by Equations (22–24). As the AFM tip starts to create a dent at the top of the bubble, the resulting additional deformation of graphene creates a pressure in the opposite direction that partially compensates the pressure exerted by the tip. The sum of the two pressures is equal to the pressure from the AFM tip:

$$P_{tip} = F(\delta)/S(\delta), \tag{27}$$

where $\delta$ is the indentation depth, $F(\delta)$ the applied force and $S(\delta)$ the part of the tip surface area in direct contact with the bubble. Therefore, the vdW pressure, $P$, can be found as a difference between $P_{tip}$ measured experimentally and the elastic energy contribution that we evaluate below.

The total energy of the bubble, for an indentation $\delta$, can be written as a sum of the work done by the force $F$ and the elastic, vdW, and internal energies:

$$E_{tot} = -F\delta + E_{el}(R, \delta) + E_{vdW}(R) + E_V[V(R, \delta)], \tag{28}$$

where $E_{vdW}$ is given by Equation (1). We assume that the energy of the material inside the bubble can be written in terms of its volume, $V(R, \delta)$, only. The elastic energy can be written as

$$E_{el}(R, \delta) = E_{el}^0(R) + \delta E_{el}(R, \delta), \tag{29}$$

where $E_{el}^0(R)$ is the elastic energy before indentation.

To estimate the effect of indentation on $\delta E_{el}(R, \delta)$ for small indentations, $\delta \ll h_{\max}$, we describe graphene at the top of the bubble as an almost flat membrane under a uniform tensile stress, $\sigma$. Given the small aspect ratios of all our bubbles, $h_{\max}/R < 0.2$, this assumption is justified both for our graphene and $MoS_2$ membranes. Then the indentation deforms the bubble over a region of radius $R^\star \lesssim R$. On dimensional grounds, the elastic energy due to the indentation can be written as

$$\delta E_{el}(\delta, R) = \sigma \int_0^{R^*} 2\pi r dr \left(\frac{\partial h}{\partial r}\right)^2 = c(v)\sigma\delta^2, \tag{30}$$

where $c(v)$ is a numerical constant that depends on the Poisson ratio of the membrane, $v$. As $\delta E_{el}$ does not depend on $R^\star$ or $R$, the minimization of $E_{tot}(R, \delta)$ with respect to $R$ leads to

$$0 = \frac{\partial E_{el}^0(R)}{\partial R} + \frac{\partial E_V(V)}{\partial V}\frac{\partial V}{\partial R} + 2\pi\gamma R. \tag{31}$$

This equation defines the dependence $R(\delta)$ of the bubble radius on the indentation depth. As a hydrocarbon material inside bubbles is essentially incompressible, we assume that $V[R(\delta), \delta]$ does not change. Then, the minimization of the total energy with respect to $\delta$ yields

$$F = 2c(v)\sigma\delta, \tag{32}$$

that is, the relation between the force and the indentation depth is expected to be linear. The nonlinearity of FDCs observed experimentally (Fig. 10) arises due to a nonlinear dependence of the contact area between the AFM tip and the pressurized bubble on $\delta$.

The value of $\sigma$ is determined by $Y$ and the strain $\epsilon$ that scales as

$$\epsilon \propto \frac{h_{\max}^2}{R^2}, \tag{33}$$

which leads to

$$\frac{F}{\delta} = d(v)Y\frac{h_{\max}^2}{R^2}, \tag{34}$$

where $d(v)$ is another dimensionless constant that depends on the Poisson ratio. If the force is applied over a finite area, defined by a contact radius $R_{contact}$, the value of $d$ in Equation (34) also depends on the ratio $R_{contact}/R$.

The vertical pressure due to the elastic deformations can be written as

$$P_{el}(r) = \frac{1}{r}\frac{\partial}{\partial r}[r\sigma_{rr}(r)\partial_r h(r)], \tag{35}$$

where $\sigma_{rr}(r)$ is the radial stress. For a flat (for example, cylindrical) tip of a radius comparable with the radius of a bubble, one would have $\partial_r h(r) = 0$ and $P_{el} = 0$, so that the pressure in the contact area would be simply $P = P_{tip} = F_{tip}/A_{tip}$, that is, the AFM tip would directly measure the pressure inside

the bubbles, as in Imbert-Fick tonometry law[34]. In our case, the tip is spherical, the contact radius, $R_{contact}$, is significantly smaller than $R$ and depends on $\delta$, which results in an additional contribution from elastic forces.

We have used Equations (26–29) to numerically fit the experimental FDCs for several graphene and $MoS_2$ bubbles, taking into account changes in $R_{contact}$ with $\delta$, as well as the increase in bubble's radius during indentation. The extracted values of vdW pressure and their dependence on $h_{max}$ and $V$ are shown in Fig. 10b,c. For $R$ in the range 250–800 nm and heights $h_{max} = 26$–115 nm, graphene and $MoS_2$ monolayers exert $P$ of the order of several MPa or tens of bar. The $P(h_{max})$ and $P(V)$ dependences obtained from experimental FDCs are in good agreement with our theory (Equations (23) and (24))—see Fig. 10b,c. The agreement implies that for graphene bubbles of a smaller height ($h_{max} \sim 1$ nm), $P$ can easily reach $\sim 100$ MPa. This is somewhat lower than the 1 GPa estimate given in ref. 12 for a hydrophobic material captured inside graphene bubbles. To this end, let us recall that $\gamma$ in Equations (23) and (24) is the difference between graphene's adhesion to the substrate, $\approx 30$ meV Å$^{-2}$ (ref. 24) and its adhesion to a material inside bubbles (Equation (1)). As graphene is lipophilic, its adhesion to hydrocarbons can be expected to be significant, thus reducing the effective $\gamma$. From our data, we extract $\gamma_G = 3.8 \pm 0.3$ meV Å$^{-2}$ and $\gamma_{MoS_2} = 6.8 \pm 0.6$ meV Å$^{-2}$ for bubbles enclosing hydrocarbons. The twice higher value of $\gamma_{MoS_2}$ compared with graphene corresponds to higher vdW pressures for the same bubble size (Figure 10 b,c).

In addition to vdW pressure, our indentation experiments allowed us to estimate the elastic stiffness (Young's moduli) of the studied 2D membranes. Figure 10d illustrates that our numerical fits to experimental FDC's are very sensitive to $Y$: changing its value in numerical fitting by 5–10% allowed us to narrow down the value of Young's modulus to $Y_G = 420 \pm 20$ N m$^{-1}$ and $Y_{MoS_2} = 210 \pm 20$ N m$^{-1}$. Both values are somewhat higher than the reported average values of $Y$ obtained using nanoindentation of suspended membranes, $350 \pm 50$ N m$^{-1}$ for graphene[35] and $180 \pm 80$ N m$^{-1}$ for $MoS_2$ (refs 17,18,28). This can be due to the fact that our 2D membranes are strained by $\sim 1\%$ due to high pressure inside. This can increase their stiffness as suggested recently[36,37].

Let us also note that, due to the high stiffness of our 2D membranes, the elastic contribution to the measured force acting on the AFM tip is comparable to that due to vdW pressure—see inset in Fig. 10d. Both pressures are approximately constant, that is, independent of the indentation depth, as expected. (The apparent variations in $P_{elastic}$ are due to the discreet nature of our numerical fitting: its value is sensitive to details of the contact between the AFM tip and bubble, which cannot be accurately reproduced at each value of $\delta$.) This implies that analysis of nanoindentation experiments in the presence of hydrostatic pressure must take into account both contributions, as done in our work.

## Discussion

We have shown that bubbles formed by monolayers of graphene, hBN and $MoS_2$ deposited onto atomically flat substrates exhibit a universal behaviour determined purely by elastic properties of the 2D crystals and independent of the properties of the trapped material.

Bubbles with smooth shapes exhibit the same aspect ratio, $h_{max}/L_{eff} \sim 0.1$, independent of their size, where $h_{max}$ is the height of a bubble and $L_{eff}$ is the characteristic length scale that describes its base. For round bubbles $L_{eff} = R$ and for triangular ones $L_{eff} \sim L$, and the distribution of values of $h_{max}/L_{eff}$ is quite narrow.

The average strain in the 2D crystal enveloping such bubbles is $\sim (h_{max}/L_{eff})^2 \approx 10^{-2}$.

Our scaling analysis shows that the value of $h_{max}/L_{eff}$ is determined by the competition between vdW adhesion and elastic energies. The vdW contribution favours formation of bubbles with a small base, and the elastic energy tends to minimize their height. While $h_{max}/L_{eff}$ does not depend on the bubble size, it depends on whether or not any residual strains in the 2D crystals are present (for example, unrelaxed strains introduced during fabrication). The remnant strain contributes mostly to the shape of the smallest bubbles, with $L_{eff} < 50$ nm.

Using AFM indentation, we were able to measure the vdW pressure exerted by graphene and $MoS_2$ membranes on the trapped material, and to extract values of vdW adhesion and Young's moduli. The pressure can be approximated by $P \sim \gamma/L_{eff} \lesssim \gamma/h_{max}$. For the relatively large bubbles in our experiments ($R = 250$–800 nm), the measured vdW pressures were in the range 1.5–4 MPa, an order of magnitude lower than could be expected from the known adhesion energy between graphene and $SiO_x$. This is attributed to non-negligible adhesion between graphene and enclosed hydrocarbons. In situations where the adhesion between graphene and trapped substances is weak, as in the case of trapped water, the vdW pressure is expected to be much higher[12]. It would be particularly interesting to measure the vdW pressure in true-nanoscale bubbles with $h \sim 1$ nm, which were not accessible in our experiments, but can exhibit pressures of the order of 1 GPa.

The combination of topographic and indentation experiments on graphene and $MoS_2$ bubbles provides an excellent method to determine the materials' elastic properties.

## Methods

**Experimental details.** The AFM images and FDCs[38] were obtained using Bruker Dimension FastScan AFM. The aspect ratio of the bubbles was measured in non-contact mode using soft cantilevers (nominal spring constant $k = 0.7$ N m$^{-1}$; nominal tip radius $r = 2$ nm) to minimize tip–sample interaction and avoid modifying the shape of the bubbles.

Indentation experiments on both graphene and $MoS_2$ bubbles were performed using silicon probes with $k = 200$ N m$^{-1}$ and $r = 8$ nm. To increase the contact area between the tip and pressurized bubbles, the cantilevers were further annealed in air at 1,000 °C for 2 h (refs 39,40). This treatment increased $r$ from the nominal 8 to $\sim 100$ nm and imparted a smooth spherical shape, as shown by the scanning electron microscopy image in Supplementary Fig. 2a. Furthermore, the shape and size of the tips used in the AFM measurements were determined via three-dimensional imaging using Bruker's tip qualification procedure on a rough Ti sample[41] (QNM kit)—see Supplementary Fig. 2b. This allowed us to find the tip radius, $R_{tip}$, for each value of $\delta$ and calculate the contact area, $S$, using a spherical tip approximation:

$$S = \pi \left( \delta^2 + R_{tip}^2 \right) \qquad (36)$$

Different probes with similar $k = 125$ and 127 N m$^{-1}$ (found using Sader's method[42]) were employed in measurements of several bubbles, and the results were independent of the probe within our experimental error. We also verified that the shape and size of the AFM tip did not change during the indentation experiments by repeatedly checking it before and after measurements.

The FDCs were obtained at the centres of graphene and $MoS_2$ bubbles. To calibrate the cantilever deflection[38,41], we used a non-deforming substrate (sapphire) as a reference. This allowed us to obtain the deflection sensitivity[38] and convert the measured signal into the force, $F$. The resulting curves ($F(d)$, where the distance $d$ is the sum of the piezoelectric actuator displacement and the cantilever deflection) are shown in Supplementary Fig. 3. Here, the indentation $\delta$ is defined as the distance $d$ such that $\delta = 0$ corresponds to zero force[43]. Accordingly, only the parts of FDCs corresponding to positive forces were analysed, as shown in Fig. 10a. As a further check that zero-indentation points were identified correctly, several bubbles were indented to their full height, that is, until the AFM tip reached the substrate (Supplementary Fig. 3b). This showed an accurate agreement between the indentation range defined above and the height of the bubble found in the scanning mode before indentation.

Repeated loading/unloading cycles on the same bubble showed high reproducibility, reversible behaviour and no signatures of fatigue (Supplementary Fig. 3a). For all the pressure measurements shown in Fig. 10, we used a loading/unloading rate of 20 nm s$^{-1}$.

**Data availability.** The data that support the findings of this study are available from the corresponding author upon request.

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

## Acknowledgements

We acknowledge funding from EPSRC and the European Commission under the Graphene Flagship, contract CNECTICT-604391. F.G. is partially funded by ERC, grant 290846 and MINECO (Spain), grant FIS2014-57432. E.K. acknowledges funding by the University of Manchester President's Doctoral Scholar Award.

## Author contributions

A.K.G, I.V.G. and F.G. designed and directed the project; E.K. performed all the AFM measurements and data analysis, including nanoindentation experiments with the help of L.F.; F.G. performed the theoretical analysis and developed models for numerical fitting of nanoindentation curves; and I.V.G. and F.G. wrote the paper with contributions from A.K.G. All authors contributed to discussions.

## Additional information

**Competing financial interests:** The authors declare no competing financial interests.

