## [Peer review file · Nature Communications]

Reviewers' comments:

Reviewer #1 (Remarks to the Author):

In this paper, the authors measure nanobubbles in 2D materials and observe universal scaling laws relating the shape of the bubbles for many different sizes. They then provide detailed calculations to explain the scaling behavior and extract the mechanical properties of the 2D materials and adhesion at the interface. This paper is a perfect example of an elegant and simple experiment that provides a large amount of insight into the properties of nanoscale materials, and is well deserving of publication. The experimental methods are carefully carried out, and the analysis is thorough. My comments are mainly on presentation with a few experimental questions:

1. The authors mention that there is evidence of different shapes and sizes for bubbles. It would be helpful to know if they different shapes appear in the same samples or in different samples, since that provides information about the underlying mechanisms of formation. If different shapes appear on the same sample, what is the relative frequency? If different shapes always appear on different samples, what is the underlying difference between the samples?
2. Along those lines, many of the fundamental lessons learned through the analysis are lost in the details. The authors should make an effort to clearly state the fundamental takeaways from all of their experiments. For example, even though they set up a mystery at the beginning of the paper, they never clearly state why the different shapes form and what is fundamentally different about them. Try to work on bringing this and other big lessons to more prominence.
3. I would suggest the authors consider another tact for figuring out additional takeaway messages as well which will broaden their impact. Imagine you are a graduate student preparing samples for electrical transport, and don't care about the mechanics of 2D materials. Using the insights gained in this paper, what can this student learn about their sample by looking at the shape, size, and distribution of bubbles to inform on their device fabrication methods and expected properties? Put these fundamental takeaways up front. "Through the experiments and analysis below, we found..."

Reviewer #2 (Remarks to the Author):

NCOMMS-16-06715

Graphene Bubbles on a Substrate: Universal Shape and Van Der Waals Pressure Khestanova et al. The authors provide a very interesting set of experiments and data on bubbles that are formed between graphene and other 2D materials. The measurements of the central heights and planform shapes of the bubbles are explained and modelled as the competition between van der Waals interactions between the membranes and the substrate and the elasticity of the membranes. Nanoindentation experiments using AFM are then used, with associated analysis, to determine the pressures inside the bubbles. The paper is very well written, has interesting results and should be published in the Journal. The following comments and annotations on the original version of the manuscript should be considered in preparing the final version.

1. Equilibrium conditions (p. 1). The authors indicate that "graphene, hBN and MoS₂ monolayers onto hBN, graphite, and MoS₂ substrates. This resulted in spontaneous formation of a large number of bubbles filled with hydrocarbons, with typical separations from ~ 0.5 to tens of microns. To ensure that the prepared hetero-structures reach equilibrium conditions, they were annealed at 150 {degree sign}C for 20-30 minutes." What is meant by equilibrium and why would the annealing treatment achieve this? What is being annealed, the membranes themselves, the hydrocarbons, etc.? What is left inside the bubbles after annealing? How was the equilibrium state actually quantified? This section is a bit vague and should be sharpened.
2. Universal profile (Fig. 6). The universal profile and fixed aspect ratio that arise from the competition between van der Waals interactions and the elasticity of the membrane are not surprising from a mechanics perspective. In fact the type of universal bubble profiles shown in Figure 6 have already been obtained in [1, 2]. It would be interesting to compare the approaches. The authors might also consider changing the title to reflect the more novel aspects of the work.
3. Analysis of the nanoindentation of a membrane. The authors might be interested in the paper by Vela et al. [3].
4. Force profiles (Fig. 10). The force profile in Figure does not show any signs of adhesive interactions between the AFM tip and membrane. These do appear in the Fig. S2 and raise the question as to the extent of their influence and why they were ignored in the analysis. Such interactions were important in some recent nanoindentation experiments on graphene on silicon [4].

1. Wang, P., et al., Numerical analysis of circular graphene bubbles. *Journal of Applied Mechanics*, 2013. 80: p. 040905.
2. Yue, K., et al., Analytical methods for the mechanics of graphene bubbles. *Journal of Applied Physics*, 2012. 112(8): p. 083512.
3. Vella, D., et al., The indentation of pressurized elastic shells: from polymeric capsules to yeast cells. *Journal of The Royal Society Interface*, 2012. 9(68): p. 448-455.
4. Suk, J.W., et al., Probing the adhesion interactions of graphene on silicon oxide by nanoindentation. *Carbon*, 2016. 103: p. 63-72.

Reply to reviewers

We wish to thank both reviewers for their favourable assessment of our manuscript and for their helpful and constructive comments.

Reviewer #1 (Remarks to the Author):

In this paper, the authors measure nanobubbles in 2D materials and observe universal scaling laws relating the shape of the bubbles for many different sizes. They then provide detailed calculations to explain the scaling behavior and extract the mechanical properties of the 2D materials and adhesion at the interface. This paper is a perfect example of an elegant and simple experiment that provides a large amount of insight into the properties of nanoscale materials, and is well deserving of publication. The experimental methods are carefully carried out, and the analysis is thorough. My comments are mainly on presentation with a few experimental questions:

1. The authors mention that there is evidence of different shapes and sizes for bubbles. It would be helpful to know if they different shapes appear in the same samples or in different samples, since that provides information about the underlying mechanisms of formation. If different shapes appear on the same sample, what is the relative frequency? If different shapes always appear on different samples, what is the underlying difference between the samples?

We thank the reviewer for this comment and apologise for not making this aspect of the experiment sufficiently clear. Bubbles of different shapes were found on each sample but not with the same frequency and there was a correlation between the bubbles' shapes and their sizes. For example, all different bubble shapes (round, triangular, pyramidal) were found on the same sample of graphene on an hBN substrate. Of these, bubbles smaller than 400 nm in radius were round or nearly round and most of them were smaller than 200 nm; bubbles with radii $500 < R < 1000$ nm were triangular with smooth tops, and triangular and pyramidal bubbles with sharp features were very few, with a broad distribution of sizes, from 400 to 1400 nm. For monolayer hBN, bubbles of all shapes tended to be smaller; accordingly, the size ranges were different (~20 to 100 nm for round bubbles; 150 to 350 nm for triangular with smooth tops) but a correlation between shape and sizes was found as well.

We have added frequency analysis for different monolayer/substrate combinations in the revised manuscript, as suggested by the reviewer (new Supplementary Figure S1), and also clarified the above correlations in the text.

2. Along those lines, many of the fundamental lessons learned through the analysis are lost in the details. The

authors should make an effort to clearly state the fundamental takeaways from all of their experiments. For example, even though they set up a mystery at the beginning of the paper, they never clearly state why the different shapes form and what is fundamentally different about them. Try to work on bringing this and other big lessons to more prominence.

We thank the reviewer for these suggestions and have incorporated a new summary discussion at the end of ‘Scaling Analysis’ section (section ‘Deviations from scaling’ has been made a subsection of ‘Scaling analysis’ in the revised manuscript, in order to emphasise the most important takeaways from our results, as suggested by the reviewer). The following summary discussion has been added:

‘The above analysis allows us to draw important conclusions about the shapes and sizes of the bubbles versus the rigidity of 2D membranes and their adhesion to the substrates. The highest order term in the bubble’s energy, $\propto h_{\max}^4 / R^2$ (equation (4)) is the elastic energy due to the deformation of the 2D crystal, proportional to the Young’s modulus, Y . Therefore, on average, softer membranes, such as monolayer MoS_2 , form more bubbles compared to e.g. graphene (cf. Figs. 2a and 2b) and most of them have round or nearly round bases. Large bubbles made by stiffer graphene ($R > 400$ nm) tend to be either triangular or pyramidal because formation of ridges associated with such shapes allows some relaxation of strain and therefore reduction in elastic energy. Accordingly, all graphene bubbles with $R > 350$ nm are either triangular or pyramidal, in contrast to MoS_2 bubbles of similar size that are (nearly) round.

Adhesion energy appears to play a smaller role in bubble shapes. Nevertheless, bubbles formed by a relatively soft membrane, such as MoS_2 , are notably higher on a substrate to which they have greater adhesion (MoS_2 on MoS_2), due to the associated higher pressure inside the bubbles (equation (11)).

Finally, there is a correlation between the shape and size of the bubbles and the presence of any other stresses in the 2D membranes, for example, in the vicinity of folds or atomic-scale steps on the substrate. As stresses associated with each bubble are not limited to its visible raised part but extend to distances $>2R$ from the bubble centre (Fig. 6a), larger bubbles are attracted to areas of stress concentration and therefore, found more often near folds, wrinkles or steps.’

3. I would suggest the authors consider another tact for figuring out additional takeaway messages as well which will broaden their impact. Imagine you are a graduate student preparing samples for electrical transport, and don't care about the mechanics of 2D materials. Using the insights gained in this paper, what can this student learn about their sample by looking at the shape, size, and distribution of bubbles to inform on their device fabrication methods and expected properties? Put these fundamental takeaways up front. "Through the experiments and analysis below, we found..."

We are grateful to the reviewer for this suggestion and have added a ‘takeaway message’ at the end of Introduction. We hope this now sets the scene and will help the reader to see the importance of bubbles not only for their mechanical properties but more generally for their role as indicators in van der Waals heterostructures. The following passage has been added:

‘Through the experiments and analysis below, we found that the in-plane stiffness of 2D crystals plays a major role in determining characteristic shapes and density of the bubbles one can expect to find when such a crystal is part of a vdW heterostructure. Stiffer 2D crystals, such as graphene or monolayer hBN on an hBN substrate, form smaller, more sparsely distributed bubbles, so that large

(up to $100 \mu\text{m}^2$) areas of the structure present a perfect vdW interface. This has been exploited in fabrication of high-quality electronic devices. On the other hand, stronger adhesion between a 2D crystal and the substrate (monolayer MoS_2 on an MoS_2 substrate being an example) can be exploited to achieve a higher vdW pressure inside the bubbles, which is desirable if one wants to modify the properties of a materials through nanoscale confinement .'

Reviewer #2 (Remarks to the Author):

NCOMMS-16-06715

Graphene Bubbles on a Substrate: Universal Shape and Van Der Waals Pressure Khestanova et al.

The authors provide a very interesting set of experiments and data on bubbles that are formed between graphene and other 2D materials. The measurements of the central heights and planform shapes of the bubbles are explained and modelled as the competition between van der Waals interactions between the membranes and the substrate and the elasticity of the membranes. Nanoindentation experiments using AFM are then used, with associated analysis, to determine the pressures inside the bubbles. The paper is very well written, has interesting results and should be published in the Journal. The following comments and annotations on the original version of the manuscript should be considered in preparing the final version.

1. Equilibrium conditions (p. 1). The authors indicate that "graphene, hBN and MoS2 monolayers onto hBN, graphite, and MoS2 substrates. This resulted in spontaneous formation of a large number of bubbles filled with hydrocarbons, with typical separations from ~ 0.5 to tens of microns. To ensure that the prepared heterostructures reach equilibrium conditions, they were annealed at $150 \text{ }^\circ\text{C}$ for 20-30 minutes." What is meant by equilibrium and why would the annealing treatment achieve this? What is being annealed, the membranes themselves, the hydrocarbons, etc.? What is left inside the bubbles after annealing? How was the equilibrium state actually quantified? This section is a bit vague and should be sharpened.

We apologise for not being sufficiently clear and thank the reviewer for pointing this out. We have added a more detailed description of sample preparation in the revised manuscript. Our samples were prepared by the so-called dry peel technique [C.P. Dean et al, Boron nitride substrates for high-quality graphene electronics, *Nature Nanotech.* **5**, 722–726 (2010); A.V. Kretinin et al. Electronic properties of graphene encapsulated with different two-dimensional atomic crystals. *Nano Lett.* **14**, 3270–3276 (2014)]. To this end graphene/monolayer hBN/monolayer MoS_2 were first mechanically exfoliated onto a polymer (PMMA) membrane. The latter was then loaded into a micromanipulator where it was placed 'face-down' onto the substrate (a thick crystal of graphite, hBN or MoS_2), after which the supporting polymer membrane was mechanically peeled off, ensuring residue-free surface of a 2D crystal. Although a few bubbles between the 2D crystal and the substrate usually form already at this stage, the 2D crystal-substrate sandwich is known to require moderate heating (annealing) to allow the hydrocarbons (that are always present on free-standing surfaces of 2D crystals) to coagulate into bubbles, leaving perfectly clean interfacial areas in between [Haigh, S. J. et al. Cross-sectional imaging of individual layers and buried interfaces of graphene-based heterostructures and superlattices. *Nat. Mater.* **11**, 764–7 (2012)].

We first ran a number of preliminary tests, where the 2D crystal-substrate structures were heated at temperatures between 130 and 150°C (known from experience) for different amounts of time. These showed that a stable arrangement of bubbles formed after the first 20-30 min at 150°C (time depending on the particular 2D crystal-substrate combination). No changes in either shape, size or position of the bubbles could be detected after that time, indicating that a stable, equilibrium structure had been achieved. Concerning the annealing temperature, we also tested heating at T

between 150 and 300°C – this did not affect either the shape or size of graphene bubbles, but changed their position; as for MoS₂ bubbles, these quickly degraded at T >150°C. Therefore for main experiments the annealing temperature of 150°C was used, which is also known to be optimum for the highest mobility graphene devices (for higher T the quality is known to deteriorate).

2. Universal profile (Fig. 6). The universal profile and fixed aspect ratio that arise from the competition between van der Waals interactions and the elasticity of the membrane are not surprising from a mechanics perspective. In fact the type of universal bubble profiles shown in Figure 6 have already been obtained in [1, 2]. It would be interesting to compare the approaches. The authors might also consider changing the title to reflect the more novel aspects of the work.

1. Wang, P., et al., Numerical analysis of circular graphene bubbles. Journal of Applied Mechanics, 2013. 80: p. 040905.

2. Yue, K., et al., Analytical methods for the mechanics of graphene bubbles. Journal of Applied Physics, 2012. 112(8): p. 083512.

We agree with the reviewer that the universal profile of pressurised membranes and the aspect ratio fixed by adhesion/in-plane stiffness are, in principle, not surprising (all can be traced to the classical theory of elasticity) and have been discussed in refs. [1,2]. We apologise for not citing these references, they have been included in the revised manuscript. Note however that, unlike those studies that analysed gas-filled bubbles under constant pressure with clamped edges, our theory corresponds to the problem studied experimentally, i.e., bubbles of a constant volume, where the edges adapt to the competition between the van der Waals attraction and the internal pressure, while the pressure itself is determined by the adhesion between the 2D crystal and the substrate. Furthermore, the 2D crystal (graphene) is free to adapt to the substrate and the bubble profiles are not assumed (as in the ref. [2]) but found self-consistently. We have added a short paragraph comparing the approaches in the revised manuscript, as suggested by the reviewer.

We have also considered revising the title of the paper, but decided against it: the central message of the paper is the experimental demonstration of the universal shape of the bubbles and measurements of van der Waals pressure (neither of which has been done before) and we believe this is best reflected in the original title.

3. Analysis of the nanoindentation of a membrane. The authors might be interested in the paper by Vella et al. [3].

[3]. Vella, D., et al., The indentation of pressurized elastic shells: from polymeric capsules to yeast cells. Journal of The Royal Society Interface, 2012. 9(68): p. 448-455.

We thank the reviewer for pointing out this paper that we were unaware of. It considers a similar problem and we now refer to it in the revised manuscript. Vella et al studied a thick elastic shell under constant pressure and a point indentation force, both of which are different from our experimental situation (see our reply to the previous point). Also neither of the derived asymptotics (equation 5.1 in Vella et al) are directly relevant to our experiments. That said, we agree with the reviewer that it is important to compare the approaches used in our study and in the paper by Vella et al and we have done this in the revised manuscript.

4. Force profiles (Fig. 10). The force profile in Figure does not show any signs of adhesive interactions between the AFM tip and membrane. These do appear in the Fig. S2 and raise the question as to the extent of their

influence and why they were ignored in the analysis. Such interactions were important in some recent nanoindentation experiments on graphene on silicon [4].

[4]. Suk, J.W., et al., Probing the adhesion interactions of graphene on silicon oxide by nanoindentation. Carbon, 2016. 103: p. 63-72.

We are sorry for not explaining properly which part of raw force-displacement curves was analysed to extract the vdW pressure and thank the reviewer for pointing this out. The force-indentation curves in Fig.10 do not include the adhesion-related 'jump-to-contact' part of the full curves, such as the one given in Supplementary Fig. S2 (now Supplementary Figure S3). As the reviewer pointed out, the 'jump-to-contact' feature corresponding to a negative force is always seen in nanoindentation experiments. However, this feature is not part of the indentation process; the latter starts from zero indentation defined as the position of the AFM tip corresponding to zero force. Accordingly, only the part of force-indentation curves corresponding to positive forces is analysed in our work. This approach is similar to that in other nanoindentation experiments (e.g. Lopez-Polin et al. Nat. Physics 2014). In the revised manuscript we have added a new Supplementary Figure S3b that clearly identifies the zero-force/zero-indentation point. Furthermore, the 'jump-to-contact' feature for our bubble indentation is smaller than usual, including the above ref. [4]. This is expected due to (i) the curved surface of the bubble and (ii) hydrophobic graphite vs say SiO_x leads to less capillary condensation.

As a further check that we have correctly identified the zero-indentation point, several bubbles were indented to their full height, i.e., until the AFM tip reached the substrate (this manifests itself as the vertical region on force-displacement curves at large load forces ~500-700nN). This showed an accurate agreement between the indentation range defined above and the height of the bubble found in the scanning mode prior to indentation. An example of such measurements is now added to the revised manuscript (Supplementary Fig. S3b).

REVIEWERS' COMMENTS:

Reviewer #1 (Remarks to the Author):

The authors have sufficiently addressed my suggestions. The article is ready for publication.

Reviewer #2 (Remarks to the Author):

The revisions made by the authors are satisfactory and I recommend that the paper should now be accepted.